# On the Utility of Koopman Operator Theory in Learning Dexterous Manipulation Skills

**Yunhai Han**[1]**, Madie Xie**[1]**, Ye Zhao**[1]**, Harish Ravichandar**[1]
[1]Georgia Institute of Technology
{yhan389, manxie, yezhao, harish.ravichandar}@gatech.edu

**Abstract:** Despite impressive dexterous manipulation capabilities enabled by learning-based approaches, we are yet to witness widespread adoption beyond well-resourced laboratories. This is likely due to practical limitations, such as significant computational burden, inscrutable learned behaviors, sensitivity to initialization, and the considerable technical expertise required for implementation. In this work, we investigate the utility of Koopman operator theory in alleviating these limitations. Koopman operators are simple yet powerful control-theoretic structures to represent complex *nonlinear* dynamics as *linear* systems in higher dimensions. Motivated by the fact that complex nonlinear dynamics underlie dexterous manipulation, we develop a Koopman operator-based imitation learning framework to learn the desired motions of both the robotic hand and the object simultaneously. We show that Koopman operators are surprisingly effective for dexterous manipulation and offer a number of unique benefits. Notably, policies can be learned *analytically*, drastically reducing computation burden and eliminating sensitivity to initialization and the need for painstaking hyperparameter optimization. Our experiments reveal that a Koopman operator-based approach can perform comparably to state-of-the-art imitation learning algorithms in terms of success rate and sample efficiency, while being *an order of magnitude* faster. Policy videos can be viewed at https://sites.google.com/view/kodex-corl.

**Keywords:** Koopman Operator, Dexterous Manipulation

## 1 Introduction

Autonomous dexterous manipulation skills are necessary for robots to successfully operate in a physical world built by and for humans. However, achieving reliable robotic dexterous manipulation has been a long-standing challenge [1] due to numerous factors, such as complex nonlinear dynamics, high-dimensional action spaces, and the expertise required to design bespoke controllers.

Over the past decade, learning-based solutions have emerged as promising solutions that can address the challenges in acquiring dexterous manipulation skills. Indeed, these methods have been shown to be capable of impressive feats, such as solving Rubik's cubes [2], manipulating Baoding balls [3], retrieving tool trays [4], and reorienting complex objects [5, 6]. However, existing learning approaches suffer from practical limitations that hinder their widespread adoption. First, implementing existing algorithms requires significant technical expertise and familiarity with modern machine learning infrastructure (e.g., knowledge of complex learning algorithms and well-established deep learning software frameworks). Second, training policies consume significant computational resources. Third, while existing approaches have achieved impressive SOTA performance, these results tend to require painstaking efforts to tune hyperparameters and architectures. Fourth, performance tends to be highly sensitive to parameter initialization.

In this work, we investigate the utility of Koopman operator theory in alleviating the limitations of existing learning-based approaches as identified above. The Koopman operator theory helps represent arbitrary *nonlinear* dynamics in finite dimensional spaces as *linear* dynamics in an infinite-dimensional Hilbert space [7]. While this equivalence is exact and fascinating from a theoretical standpoint, it is not tractable. However, recent advances have enabled the approximation of this equivalence in higher but *finite*-dimensional spaces by learning the operator directly from data [8].

7th Conference on Robot Learning (CoRL 2023), Atlanta, USA.

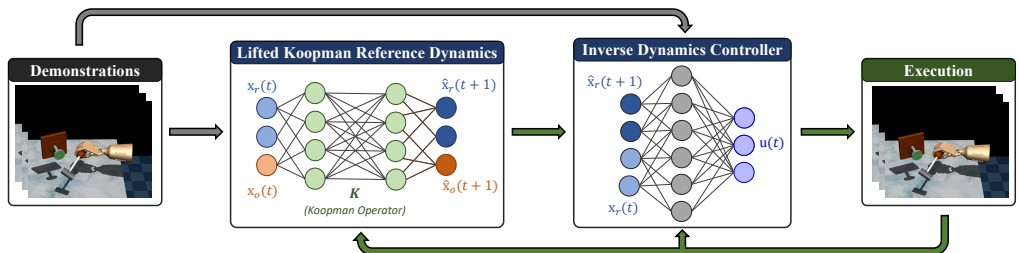

Figure 1: KODex simultaneously encodes complex nonlinear dynamics of the desired motion of both the robot state ($x_r$) and the object state ($x_o$) as a *linear* dynamical system in a higher-dimensional space by learning a Koopman operator $\boldsymbol{K}$ directly from demonstrations. Further, KODex learns an inverse dynamics controller to track the robot reference trajectory ($\{\hat{x}_r(t)\}_{t=1}^{T}$) generated by the lifted linear system.

We develop a novel imitation learning framework, dubbed *Koopman Operator-based Dexterous Manipulation (KODex)*, to evaluate the utility of Koopman operator theory for dexterous manipulation (see Fig. 1). Specifically, we model desired behaviors as solutions to nonlinear dynamical systems and learn Koopman operators that define approximately-equivalent linear dynamics in higher-dimensional spaces. Note that it is insufficient to exclusively focus on the robot's motion as the objective of dexterous manipulation is centered on the object's motion [1]. As such, KODex *simultaneously* learns the desired motions of both the robot and the object from demonstrations. To eliminate the need for an expertly-tuned controller, KODex utilizes a learned inverse dynamics controller to track the reference trajectory generated by the learned dynamical system.

A significant benefit of learning Koopman operators from data is that it lends itself to an *analytical* solution. As such, KODex is simple to implement and does not require expertise and familiarity with state-of-the-art (SOTA) machine learning infrastructure. Instead of painstaking hyperparameter optimization, we show that generic and task-agnostic polynomial lifting functions are sufficient for KODex to learn diverse dexterous manipulation skills. Further, KODex offers consistent and predictable performance since the learning process is analytical and thus not sensitive to parameter initialization. Finally, given that KODex learns a *linear* dynamical system, one can readily inspect the learned behaviors using a wide array of control-theoretic tools.

We carry out extensive evaluations of KODex within the context of four dexterous manipulation skills on the simulated Adroit hand, an established experimental platform for dexterous manipulation [9]. Further, we compare KODex against SOTA imitation learning approaches in terms of general efficacy, computational efficiency, and sample efficiency. Our results demonstrate that KODex is at least an *order of magnitude* faster to train than SOTA imitation learning algorithms, while achieving comparable sample efficiency and task success rates. These results suggest that Koopman operators can be effective, efficient, and reliable tools to learn dexterous manipulation skills and to reduce the barriers to wide-spread adoption.

## 2   Related Work

In this section, we contextualize our contributions within relevant sub-fields.

**Learning Manipulation Skills as Dynamical Systems**: Our work falls into the category of dynamical-system-based imitation learning methods for manipulation [10]. Made popular by the Dynamics Movement Primitives (DMPs) [11], these methods model robot motions as solutions to a learnable dynamical system. The past decade witnessed a plethora of approaches built upon the same principle (e.g., [12–17]), creating increasingly-capable LfD tools for manipulation. Robustness to perturbations, high sample efficiency, and provable convergence are all but a few examples of the many advantages of dynamical-system-based approaches. These approaches tend to be highly structured and leverage control-theoretic and topological tools to learn complex desired motions with unparalleled sample efficiency. Recent work also embedded the dynamical systems structure into deep neural networks to enable end-to-end learning [18]. These approaches were primarily designed to capture low-dimensional end-effector skills for serial-link manipulators. In contrast, our work investigates the utility of Koopman operators in learning dexterous manipulation skills on high-DOF platforms.

**Learning Dexterous Manipulation Skills**: Deep Reinforcement Learning (RL) has been dominating the field of dexterous manipulation recently, enabling an impressive array of skills [3, 19, 20]. A popular approach demonstrated that a multi-finger hand can learn to solve the Rubik's cube OpenAI et al. [2]. Recently, Chen et al. [5] developed a model-free RL framework capable of reorienting over 2000 differently-shaped objects, and Khandate et al. [6] combined sampling-based planning and model-free RL methods on the same task. Despite impressive successes, RL-based algorithms suffer from poor sample efficiency and notoriously-difficult training procedures. In contrast, Imitation learning (IL) aims to improve sample efficiency by leveraging expert demonstrations [10, 21]. However, most existing IL-based methods (including those discussed in Section 2) focus on lower-DOF manipulators and do not scale well to high-dimensional systems. Indeed, there are at least two recent notable exceptions to this limitation: Xie et al. [4] developed a highly structured IL method to learn dexterous manipulation skills from demonstrations in the form of a dynamical system; Arunachalam et al. [22] introduced a mixed-reality framework to collect high-quality demonstrations and learn dexterous manipulation skills by leveraging visual representations and motion retargeting. Researchers have also combined IL with RL to get the best of both approaches and has been able to achieve impressive performance (e.g., [9, 23]). A common attribute of all existing learning approaches to dexterous manipulation is that they rely on significant computational resources and user expertise for implementation and hyperparameter tuning. Further, the effectiveness of these approaches is highly sensitive to parameter initialization [24]. In stark contrast, KODex encodes complex skills as dynamical systems which are *analytically* extracted from demonstrations. As such, KODex incurs a significantly smaller computational burden and eliminates the dependence on painstaking hyperparameter tuning and unreliable numerical optimization. Further, unlike opaque deep neural networks, KODex learns *linear* dynamical systems that can be inspected via control-theoretic tools.

**Koopman Operators in Robotics**: Recently, Koopman operator theory has proven beneficial in various robotic systems, such as differential drive robots [25], spherical and serial-link manipulators [26], autonomous excavators [27, 28], and soft robotic manipulators [29, 30]. However, the systems investigated in these works are low-dimensional. In contrast, our work is focused on evaluating the effectiveness of Koopman operators in learning skills for a high-dimensional system with complex dynamics (i.e., a multi-fingered hand). Further, prior works have not sufficiently investigated the relative benefits of leveraging Koopman operators over SOTA neural network-based approaches, and the circumstances under which these benefits hold. In our work, we thoroughly evaluate KODex against SOTA imitation learning methods within the context of multiple dexterous manipulation tasks.

## 3 Preliminary: Koopman Operator Theory

We begin by providing a brief introduction to Koopman operator theory [7].

**Koopman Representation**: Consider a discrete-time autonomous nonlinear dynamical system

$$\mathrm{x}(t+1) = F(\mathrm{x}(t)), \tag{1}$$

where $\mathrm{x}(t) \in \mathcal{X} \subset \mathbb{R}^n$ is the state at time $t$, and $F(\cdot) : \mathbb{R}^n \to \mathbb{R}^n$ is a nonlinear function.

To represent the the nonlinear dynamical system in (1) as a linear system, we begin by introducing a set of *observables* using the so-called *lifting function* $g : \mathcal{X} \to \mathcal{O}$, where $\mathcal{O}$ is the space of observables. We can now define the *Koopman Operator* $\mathcal{K}$, an infinite-dimensional operator on the lifting function $g(\cdot)$ for the discrete time system defined in (1) as follows

$$[\mathcal{K}g] = g(F(\mathrm{x}(t))) = g(\mathrm{x}(t+1)) \tag{2}$$

If the observables belong to a vector space, the Operator $\mathcal{K}$ can be seen as an infinite-dimensional linear map that describes the evolution of the observables as follows

$$g(\mathrm{x}(t+1)) = \mathcal{K}g(\mathrm{x}(t)) \tag{3}$$

Therefore, the Koopman operator $\mathcal{K}$ linearly propagates forward the infinite-dimensional lifted states (i.e., observables). In practice, we do not benefit from this representation since it is infinite-dimensional. However, we can approximate $\mathcal{K}$ using a matrix $\mathbf{K} \in \mathbb{R}^{p \times p}$ and define a finite set of observables $\phi(t) \in \mathbb{R}^p$. Thus, we can rewrite the relationship in (3) as

$$\phi(\mathrm{x}(t+1)) = \mathbf{K}\phi(\mathrm{x}(t)) + r(\mathrm{x}(t)), \tag{4}$$

where $r(\mathrm{x}(t)) \in \mathbb{R}^p$ is the residual error caused by the finite dimensional approximation, which can be arbitrarily reduced based on the choice of $p$.

**Learning Koopman Operator from Data**: The matrix operator $\mathbf{K}$ can be inferred from a dataset $D = [\mathrm{x}(1), \mathrm{x}(2), \cdots, \mathrm{x}(T)]$, which contains the solution to the system in (1) for $T$ time steps. Given the choice of observables $\phi(\cdot)$, the finite dimensional Koopman matrix $\mathbf{K}$ is computed by minimizing the approximation error defined in (4). Specifically, we can obtain $\mathbf{K}$ from $D$ by minimizing the cost function $\mathbf{J}(\mathbf{K})$ given below

$$\mathbf{J}(\mathbf{K}) = \frac{1}{2} \sum_{t=1}^{t=T-1} \|r(\mathrm{x}(t))\|^2 = \frac{1}{2} \sum_{t=1}^{t=T-1} \|\phi(\mathrm{x}(t+1)) - \mathbf{K}\phi(\mathrm{x}(t))\|^2 \tag{5}$$

Note minimizing $\mathbf{J}(\mathbf{K})$ amounts to solving a least-square problem, whose solution is given by [8]

$$\mathbf{K} = \mathbf{A}\mathbf{G}^\dagger; \ \ \mathbf{A} = \frac{1}{T-1} \sum_{t=1}^{t=T-1} \phi(\mathrm{x}(t+1)) \otimes \phi(\mathrm{x}(t)), \ \ \mathbf{G} = \frac{1}{T-1} \sum_{t=1}^{t=T-1} \phi(\mathrm{x}(t)) \otimes \phi(\mathrm{x}(t)) \tag{6}$$

where $\mathbf{G}^\dagger$ denotes the Moore–Penrose inverse[1] of $\mathbf{G}$, and $\otimes$ denotes the outer product.

# 4 Learning Koopman Operators for Dexterous Manipulation

We begin by introducing our framework to model dexterous manipulation skills as nonlinear dynamics and discuss the importance of incorporating object states into the system (Section 4.1). Next, we describe how KODex learns the reference dynamics[2] for a given skill from demonstrations (Section 4.2). Then, we discuss how to learn a low-level controller, also from demonstrations, in order to faithfully track the reference trajectories generated by KODex (Section 4.3). Finally, we discuss policy execution (Section 4.4). Further, an overall pseudo-code for KODex can be found in Appendix A.

## 4.1 Modeling Dexterous Manipulation Skills

A central principle behind KODex is that the desired behavior of a robot can be represented using a dynamical system. Note that, unlike other kinds of manipulation skills (e.g., end-effector skills of multi-link manipulators), dexterous manipulation is explicitly concerned with how an object moves as a result of the robot's motion [1]. As such, KODex captures the desired motion of the robot along with that of the object. To this end, we define the state at time $t$ as $\mathrm{x}(t) = [\mathrm{x}_r(t)^\top, \mathrm{x}_o(t)^\top]^\top$, where $\mathrm{x}_r(t) \in \mathcal{X}_r \subset \mathbb{R}^n$ and $\mathrm{x}_o(t) \in \mathcal{X}_o \subset \mathbb{R}^m$ represent the state of the robot and the object, respectively, at time $t$. As such, the dynamical system we wish to capture is

$$\mathrm{x}(t+1) = F^*(\mathrm{x}(t)) \tag{7}$$

where $F^*(\cdot) : \mathcal{X}_r \times \mathcal{X}_o \to \mathcal{X}_r \times \mathcal{X}_o$ denotes the true dynamics that govern the *interdependent* motions of the robot and the object. Note that this system is time-invariant. Indeed, time-invariant dynamical systems provide a natural way to capture manipulation skills that are more robust to intermittent perturbations than those that explicitly depend on time [10].

A key challenge in learning the dynamical system in (7) is that it can be arbitrarily complex and highly nonlinear, depending on the particular skill of interest. KODex leverages Koopman operator theory to learn a *linear* dynamical system that can effectively approximate such complex nonlinear dynamics. To this end, we first define a set of observables as follows

$$\phi(\mathrm{x}(t)) = [\mathrm{x}_r(t)^\top, \psi_r(\mathrm{x}_r(t)), \mathrm{x}_o(t)^\top, \psi_o(\mathrm{x}_o(t))]^\top, \forall t \tag{8}$$

where $\psi_r : \mathbb{R}^n \to \mathbb{R}^{n\prime}$ and $\psi_o : \mathbb{R}^m \to \mathbb{R}^{m\prime}$ are vector-valued lifting functions that transform the robot and object state respectively. While there are no coupling terms between the robot and the object states in (8), note that the robot and object states still mix after the lifting operation.

---

[1] It could be efficiently computed using the `scipy.linalg.pinv(`$\mathbf{G}$`)` function from Scipy library.

[2] We use the term "reference dynamics" to describe a fictitious dynamical system that encodes task-specific ideal trajectories in the configuration space, and not the physical robot dynamics.

In our implementation, we use polynomial functions up to a finite degree in our lifting function since polynomial functions allow for flexible definition of complex functions. However, it is important to note that KODex is agnostic to the specific choice of observables. Further, we do not assume that we know the ideal set of observables for any given skill. Instead, as we demonstrate in Section 5, KODex can learn different dexterous manipulation skills on the same space of observables.

## 4.2 Learning Reference Dynamics

We now turn to the challenge of learning the Koopman operator $\mathbf{K}$ from demonstrations. Let $D = [\{x^{(1)}(t), \tau^{(1)}(t)\}_{t=1}^{t=T^{(1)}}, \cdots, \{x^{(N)}(t), \tau^{(N)}(t)\}_{t=1}^{t=T^{(N)}}]$ denote a set of $N$ demonstrations containing trajectories of state-torque pairs. Now, we can compute the Koopman matrix as $\mathbf{K} = \mathbf{AG}^{\dagger}$, where $\mathbf{A}$ and $\mathbf{G}$ can be computed by modifying the expressions in (6) as follows

$$\mathbf{A} = \sum_{n=1}^{n=N} \sum_{t=1}^{t=T^{(n)}-1} \frac{\phi(x^n(t+1)) \otimes \phi(x^n(t))}{N(T^{(n)}-1)}, \quad \mathbf{G} = \sum_{n=1}^{n=N} \sum_{t=1}^{t=T^{(n)}-1} \frac{\phi(x^n(t)) \otimes \phi(x^n(t))}{N(T^{(n)}-1)} \quad (9)$$

It is worth noting that KODex can also leverage partial trajectories that do not complete the task, as long as all the state pairs $(x^n(t), x^n(t+1))$ are temporally consecutive. Additionally, we also record the actuated torque $\tau(t)$ at each time step for the controller design discussed in Section 4.3.

We use the learned reference dynamics to generate rollouts in the observables space. However, we need to obtain the rollouts in the original robot states to command the robot. Since we designed $\phi(x(t))$ such that both robot state $x_r(t)$ is a part of the observables in (8), we can retrieve the desired robot trajectory $\{\hat{x}_r(t)\}$ by selecting the corresponding elements in $\phi(x(t))$.

Indeed, the data distribution in $D$ has a considerable effect on the generalizability of the computed Koopman matrix $\mathbf{K}$. Therefore, there is an inevitable trade-off between the number of demonstrations and the cost of data collection - a challenge shared by most imitation learning algorithms.

## 4.3 Learning a Tracking Controller

To track the desired trajectories generated from the learned reference dynamics, we learn an inverse dynamics controller $C$ [31] [18]. Indeed, a PD controller could be designed instead of learning a tracking controller. However, one would have to painstakingly tune the control gains and frequency, and do so for each task independently.

We use a multi-layer perception (MLP) as the tracking controller and train it using the recorded state-torque pairs $(x_r^n(t), x_r^n(t+1), \tau^n(t))$ by minimizing

$$\mathcal{L}_{\text{control}} = \sum_{n=1}^{n=N} \sum_{t=1}^{t=T^{(n)}-1} \frac{|C(x_r^n(t), x_r^n(t+1)) - \tau^n(t)|^2}{N(T^{(n)}-1)} \quad (10)$$

The learned controller takes as input the current robot state $x_r(t)$ and the desired next state from the reference trajectory $\hat{x}_r(t+1)$, and generates the torque $\tau(t)$ required for actuation.

## 4.4 Execution

With the reference dynamics and the tracking control learned, we specify how to execute the policy in this section. Suppose $x(1) = (x_r(1), x_o(1))$ is the given initial state, we can generate the reference trajectory $(\{\hat{x}_r(t)\}_{t=1}^{T})$ by propagating the learned reference dynamics $\hat{x}(t+1) = \mathbf{K}\hat{x}(t)$. Further, at time step $t$, we pass the current robot state $x_r(t)$ and the desired next robot state $\hat{x}_r(t+1)$ to the learned controller $C$ to compute the required torque $\tau(t)$.

# 5 Experimental Evaluation

We evaluated KODex along with existing approaches in terms of their general efficacy, computational efficiency, sample efficiency, and scalability when learning dexterous manipulation skills.

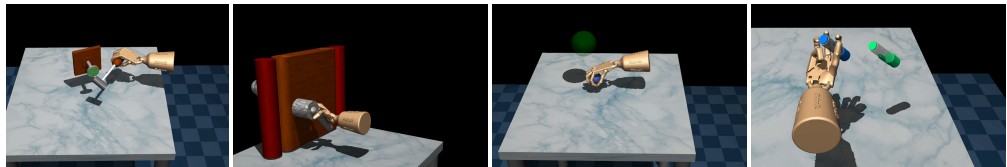

Figure 2: We evaluate KODex on four tasks from [9]: Tool Use, Door Opening, Relocation, and Reorientation.

## 5.1 Experimental Design

**Evaluation Platform**: We conducted all our experiments on the widely-used ADROIT Hand [9] – a 30-DoF simulated system (24-DoF hand + 6-DoF floating wrist base) built with MuJoCo [32].

**Baselines**: We compared KODex against the following baselines:

- *NN*: Fully-connected neural network policy
- *LSTM*: Recurrent neural network policy with Long Short-Term Memory (LSTM) units
- *NDP*: Neural Dynamic policy [18]
- *NGF*: Neural Geometric Fabrics policy [4]

Note that NN and LSTM are unstructured baselines which help question the need for structured policies. NDP and NGF are highly-structured SOTA imitation learning methods for manipulation.

We undertook several precautions to ensure a fair comparison. First, we designed the robot and object state space for all baselines and KODex to be identical. Second, we carefully designed the baselines policies and tuned their hyper-parameters for each baseline method (Appendices E and F). Third, we trained each baseline policy over five random seeds to control for initialization effects. For all tasks, we saved the baseline policies that performed the best on a validation set of 50 held-out demonstrations. Note that KODex utilizes an analytical solution and thus does not require parameter initialization or hyper parameter optimization.

**Tasks**: We evaluated all algorithms on a set of four tasks originally proposed in [9] (see Fig. 2).

- *Tool use*: Pick up the hammer to drive the nail into the board placed at a randomized height.
- *Door opening*: Given a randomized door position, undo the latch and drag the door open.
- *Object relocation*: Move the blue ball to a randomized target location (green sphere).
- *In-hand reorientation*: Reorient the blue pen to a randomized goal orientation (green pen).

See Appendix B for the state space design of all tasks. For all tasks, the reference dynamics was queried at 100HZ, and the controller ran at 500HZ.

**Metrics**: We quantify performance in terms of i) *Training time*: Time taken to train a policy, ii) *Imitation error*: The $L1$ distance between generated joint trajectories and the demonstrations, and iii) *Task success rate*: Percentage of successful trials (see Appendix C for success criteria).

**Expert Policy**: For each task, we trained an expert RL agent using DAPG [9] to generate 250 expert demonstrations (200 for training and 50 for validation). See Appendix D for further details.

**Inverse Dynamics Controller**: To standardize controller performance across methods, we trained a common inverse dynamics controller for each task using 250 demonstrations (see Appendix G).

## 5.2 General Efficacy

In Fig. 3, we report the training time, imitation error, and task success rate for each method on each task, when trained on 200 demonstrations and tested on 10,000 testing instances.

**Training time**: As can be seen, KODex is *an order of magnitude* faster than both unstructured baselines (NN, LSTM) and SOTA IL methods (NDP, NGF). Further, this trend holds across all the tasks. This is to be expected since KODex analytically computes the Koopman operator unlike all the baselines, which rely on gradient descent and numerical optimization.

**Imitation error**: In general, all methods (except NN) achieve low imitation error for the Tool Use task with negligible difference across methods. In the three remaining tasks, we see that all structured methods (NDP, NGF, KODex) considerably outperform the unstructured baseline (LSTM).

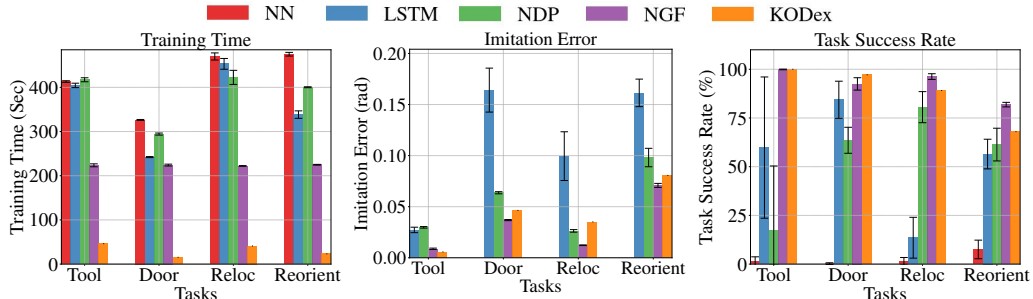

Figure 3: We report training time (left), imitation error (middle), and success rate (right) for methods on each task when trained on 200 demonstrations and evaluated on an independent set of 10,000 samples. Error bars for baseline methods, show the standard deviation over five random seeds.

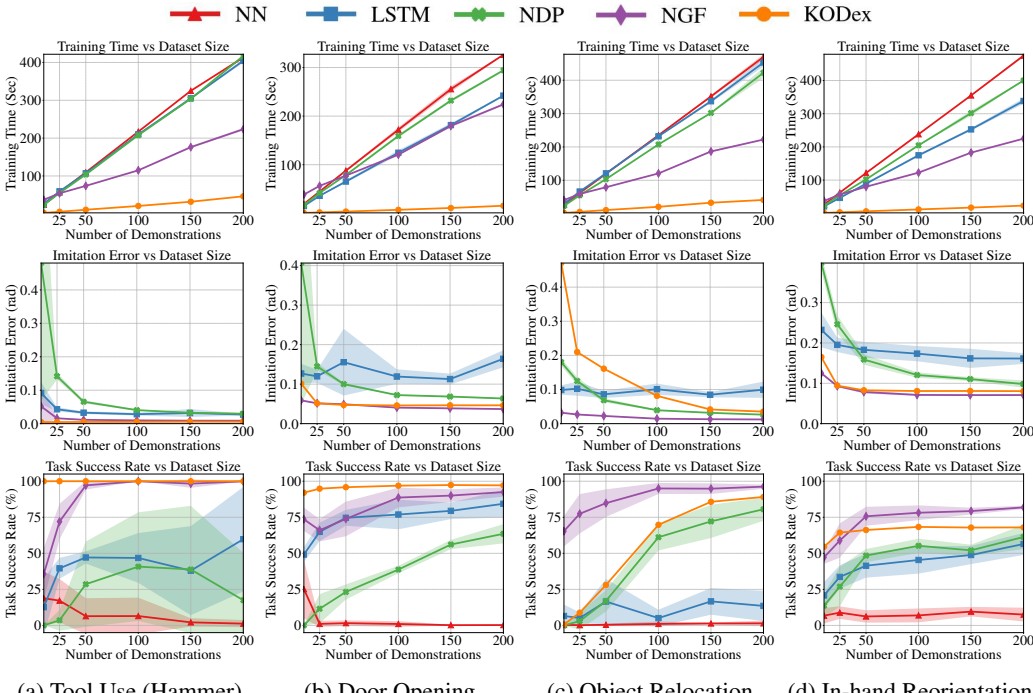

(a) Tool Use (Hammer)    (b) Door Opening    (c) Object Relocation    (d) In-hand Reorientation

Figure 4: The effects of number of demonstrations on training time (top row), imitation error (middle row), and success rate (bottom row) for all methods on each task. Solid lines indicate mean trends and shaded areas show ± standard deviation over five random seeds.

We have excluded the significantly larger imitation errors generated by NN from the plot to preserve the resolution necessary to distinguish between the other methods. These results reinforce the effectiveness of structured methods in imitating demonstrations. Importantly, KODex is able to achieve imitation performance comparable to the SOTA IL methods despite its simplicity, while remaining an order of magnitude more computationally efficient.

**Task success rate**: As expected, the NN policy performs significantly worse than all other methods. On the other hand, LSTM achieves impressive task success rates, even outperforming NDP in two of the four tasks. This is in stark contrast to its high imitation error. While counter-intuitive, this observation follows the recent finding that imitation error and task success rate might not necessarily be correlated [33]. We observe that KODex and NGF perform comparably, with one achieving a higher task success rate than the other in two of the four tasks. Importantly, KODex results in the most consistent and predictable performance due to its lack of sensitivity to initialization.

## 5.3 Scalability and Sample Efficiency

To investigate scalability and sample efficiency, we trained policies on a varying number of demonstrations ([10, 25, 50, 100, 150, 200]). In Fig. 4, we report the training time, imitation error, and task success rate for each method as a function of the number of demonstrations when tested on the same 10,000 instances used to evaluate general efficacy.

**Training time**: We observe that KODex scales with the number of demonstrations significantly better than the baselines, as evidenced by its training time growing at considerably lower rates.

**Imitation error and Success rate**: We find that unstructured models (NN and LSTM) fail to demonstrate a consistent monotonic decrease (increase) in imitation error (task success rate) as the number of demonstrations increase. In stark contrast, structured methods (NDP, NGF, and KODex) are able to consistently drive down imitation error and improve task success rate. KODex almost consistently achieves the lowest imitation error and the highest task success rate with the fewest number of demonstrations, and is closely followed by NGF. These observations suggest that KODex tends to be comparably, if not more sample efficient than the baselines, thanks to the rich structure induced by the Koopman operator and the resulting effectiveness in capturing nonlinear dynamics. The only exception to this trend is the Object Relocation task, in which KODex requires 150 demonstrations to perform comparably to NGF. We speculate this is because the demonstrations for this task exhibit high variance as the hand base moves across a large space, and KODex requires more demonstrations to capture the reference dynamics.

## 5.4 Additional Experiments

Additional experiments reported in the appendix suggest that KODex learns policies that i) have inference time on par with SOTA baselines (Appendix H), ii) have zero-shot out-of-distribution generalization comparable to SOTA IL methods (Appendix I), iii) are robust to changes in physical properties (Appendix J), iv) are not overly sensitive to the choice of basis functions (Appendix K), v) are nearly-stable linear dynamical systems that generate safe and smooth robot trajectories (Appendix L), and vi) are significantly more efficient and scalable than a baseline BC method that directly learns state-action mappings (Appendix M).

# 6 Conclusions

We investigated the utility of Koopman operator theory in learning dexterous manipulation skills by encoding complex nonlinear reference dynamics as linear dynamical systems in higher-dimensional spaces. Our investigations conclusively show that a Koopman-based framework can i) analytically learn dexterous manipulation skills, eliminating the sensitivity to initialization and reducing the need for user expertise, and ii) match or outperform SOTA imitation learning approaches on various dexterous manipulation tasks, while being an order of magnitude faster.

# 7 Limitations and Future Work

While our work offers promise for the utility of Koopman operators in dexterous manipulation, it reveals numerous avenues for further improvement. First, we did not deploy KODex on physical robots. Although our results on robustness to changes in physical properties show promise, we plan to deploy KODex on physical platforms to translate our findings to hardware. Second, we only considered polynomial basis functions; other non-smooth functions (e.g., ReLU [34]) could be beneficial to manipulation tasks involving friction and contact. Third, KODex has limited out-of-distribution generalization, much like most existing imitation learning approaches. Future work can investigate if additional data collection [35] and learned lifting functions [36–38] alleviate this concern. Fourth, KODex relies on demonstrated action trajectories to learn the tracking controller and reduce human effort. It might be possible to instead use reinforcement learning [39], thereby enabling the ability to learn from state-only observations [40]. Fifth, KODex could be evaluated on other domains and robotics tasks (e.g., the benchmark tasks in [41]) to further understand the tradeoffs between KODex and other imitation learning approaches. Finally, Koopman operators can be used to learn system dynamics via self play to enable model-based reinforcement learning.

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

# Appendices

## A  KODex Pseudo-code

The overall pseudo-code for KODex is shown below.

---
**Algorithm 1:** KODex

---
**Demonstration Data Collection**
Initialize $D = \varnothing$;
**for** $n \in \{1, ..., N\}$ **do**
    Generate a $T^{(n)}$-horizon trajectory of states and torques $\{[x^n(t), \tau^n(t)]\}_{t=1}^{t=T^{(n)}}$;
    Add $\{[x^n(t), \tau^n(t)]\}_{t=1}^{t=T^{(n)}}$ to $D$;
**end**
**Koopman Operator Approximation**
Determine lifting function $\phi(x(t))$;
Compute $\mathbf{K}$ on $D$ (6, 9);
**Controller Design**
Build a controller $C$ as a neural network with inputs as $(x_r(t), x_r(t+1))$ and output as $\tau(t)$;
Train $C$ using state-torque pairs $(x_r^n(t), x_r^n(t+1), \tau^n(t))$ in $D$ (10);
**Execution**
Specify the initial states $x(1)$;
**for** $t \in \{1, ..., T-1\}$ **do**
    Predict the next robot states $\hat{x}_r(t+1)$ using $\mathbf{K}$ (3 8);
    Read the current robot states $x_r(t)$;
    Generate the torque $\tau(t)$ using $C$ on $(x_r(t), \hat{x}_r(t+1))$ and execute it;
**end**

---

## B  State Design

In this section, we show the state design for each task in detail. It should be noted that the motion capability of the hand for each task were suggested from the work [9] that originally introduced these tasks. For a decent implementation, we employed the same setting.

**Tool use** For this task, the floating wrist base can only rotate along the $x$ and $y$ axis, so we have $x_r(t) \in \mathcal{X}_r \subset \mathbb{R}^{26}$. Regarding the object states, unlike the other tasks, where the objects of interest are directly manipulated by the hand, this task requires to modify the environment itself. As a result, except for the hammer positions, orientations and their corresponding velocities $p_t^{tool}, o_t^{tool}, \dot{p}_t^{tool}, \dot{o}_t^{tool}$ ($\mathbb{R}^3$), we also define the nail goal position $p^{nail}$ ($\mathbb{R}^3$). Finally, we have $x_o(t) = [p_t^{tool}, o_t^{tool}, \dot{p}_t^{tool}, \dot{o}_t^{tool}, p^{nail}] \in \mathcal{X}_o \subset \mathbb{R}^{15}$. As a result, $x(t)$ includes 41 states in total and we use $T = 100$.

**Door opening** For this task, the floating wrist base can only move along the direction that is perpendicular to the door plane but rotate freely, so we have $x_r(t) \in \mathcal{X}_r \subset \mathbb{R}^{28}$. Regarding the object states, we define the fixed door position $p^{door}$, which can provide with case-specific information (similar to $p^{nail}$ in Tool Use), and the handle positions $p_t^{handle}$ (both $\mathbb{R}^3$). In order to take into consideration the status of door being opened, we include the angular velocity of the opening angle $v_t$($\mathbb{R}^1$). Finally, we have $x_o(t) = [p_t^{handle}, v_t, p^{door}] \in \mathcal{X}_o \subset \mathbb{R}^7$. As a result, $x(t)$ includes 35 states in total and we use $T = 70$.

**Object relocation** For this task, the ADROIT hand is fully actuated, so we have $x_r(t) \in \mathcal{X}^r \subset \mathbb{R}^{30}$ (24-DoF hand + 6-DoF floating wrist base). Regarding the object states, we define $p^{target}$ and $p_t^{ball}$ as the target and current positions. Then, we compute $\bar{p}_t^{ball} = p_t^{ball} - p^{target}$, which is the component of $p_t^{ball}$ in a new coordinate frame that is constructed by $p^{target}$ being the origin. We additional include the ball orientation $o_t^{ball}$ and their corresponding velocities $\dot{p}_t^{ball}, \dot{o}_t^{ball}$ (all $\mathbb{R}^3$). Finally, we have $x_o(t) = [\bar{p}_t^{ball}, o_t^{ball}, \dot{p}_t^{ball}, \dot{o}_t^{ball}] \in \mathcal{X}_o \subset \mathbb{R}^{12}$. As a result, $x(t)$ includes 42 states in total and we use $T = 100$.

**In-hand reorientation** For this task, the floating wrist base is fixed, so we only consider the 24-DoF hand joints. Therefore, we have $x_r(t) \in \mathcal{X}_r \subset \mathbb{R}^{24}$. Regarding the object states, we define $o^{goal}$

and $o_t^{\text{pen}}$ as the goal and current pen orientations, which are both unit direction vectors. Then, we transform $o_t^{\text{pen}}$ to a new rotated coordinate frame that is constructed by $o^{\text{goal}}$ being $x$ axis ([1,0,0]). Note that the vector $\bar{o}_t^{\text{pen}}$ after transformation is also a unit vector and it converges to x axis if the pen is perfectly manipulated to goal orientation $o^{\text{goal}}$. In addition, we also include the center of mass position $p_t^{\text{pen}}$ and their corresponding velocities $\dot{p}_t^{\text{pen}}$, $\dot{o}_t^{\text{pen}}$ (all $\mathbb{R}^3$). Finally, we have $x_o(t) = [p_t^{\text{pen}}, \bar{o}_t^{\text{pen}}, \dot{p}_t^{\text{pen}}, \dot{o}_t^{\text{pen}}] \in \mathcal{X}_o \subset \mathbb{R}^{12}$. As a result, $x(t)$ includes 36 states in total and we use $T = 100$.

In this work, we only included the joint positions as the robot states (with the only exception of NGF's second-order policy) for the following reasons: 1) Given that these tasks are not repetitive, we found that joint position information was sufficient to disambiguate the robot's next action, 2) even when ambiguity arises for a given joint position, object state information can help with disambiguation. Further, the impressive performance achieved by KODex in our experiments support this design choice. Indeed, KODex is agnostic to this specific state design. One can incorporate velocity information into the robot state space without the need of any changes to the training procedure.

## C  Task Success Criteria

The task success criteria are listed below. The settings were the same as proposed in [9].
**Tool Use:** The task is considered successful if at last time step $T$, the Euclidean distance between the final nail position and the goal nail position is smaller than 0.01.
**Door Opening:** The task is considered successful if at last time step $T$, the door opening angle is larger than 1.35 rad.
**Object Relocation:** At each time step $t$, if $\sqrt{|p^{\text{target}} - p_t^{\text{ball}}|^2} < 0.10$, then we have $\rho(t) = 1$. The task is considered successful if $\sum_{t=1}^{T} \rho(t) > 10$.
**In-hand Reorientation:** At each time step $t$, if $o^{\text{goal}} \cdot o_t^{\text{pen}} > 0.90$ ($o^{\text{goal}} \cdot o_t^{\text{pen}}$ measures orientation similarity), then we have $\rho(t) = 1$. The task is considered successful if $\sum_{t=1}^{T} \rho(t) > 10$.

## D  Sampling Procedure

We describe the sampling procedure in this section. The sample distributions used for RL training and demo collection were identical, as suggested in [9]. The out-of-distribution data were generated to evaluate the zero-shot out-of-distribution generalizability of each policy.
**Tool Use:** We randomly sampled the nail heights ($h$) from a uniform distributions. Within distribution: we used $h \in \mathcal{H} \sim \mathcal{U}(0.1, 0.25)$; Out of distribution: we used $h \in \mathcal{H} \sim \mathcal{U}(0.05, 0.1) \cup \mathcal{U}(0.25, 0.3)$.
**Door Opening:** We randomly sampled the door positions ($xyz$) from uniform distributions. Within distribution: we used $x \in \mathcal{X} \sim \mathcal{U}(-0.3, 0)$, $y \in \mathcal{Y} \sim \mathcal{U}(0.2, 0.35)$, and $z \in \mathcal{Z} \sim \mathcal{U}(0.252, 0.402)$; Out of distribution: we used $y \in \mathcal{Y} \sim \mathcal{U}(0.15, 0.2) \cup \mathcal{U}(0.35, 0.4)$ ($x, z$ remained unchanged).
**Object Relocation:** We randomly sampled the target positions ($xyz$) from uniform distributions. Within distribution: we used $x \in \mathcal{X} \sim \mathcal{U}(-0.25, 0.25)$, $y \in \mathcal{Y} \sim \mathcal{U}(-0.25, 0.25)$, and $z \in \mathcal{Z} \sim \mathcal{U}(0.15, 0.35)$; Out of distribution: we used $z \in \mathcal{Z} \sim \mathcal{U}(0.35, 0.40)$ ($x, y$ remained unchanged).
**In-hand Reorientation:** We randomly sampled the pitch ($\alpha$) and yaw ($\beta$) angles of the goal orientation from uniform distributions. Within distribution: we used $\alpha \in \mathcal{A} \sim \mathcal{U}(-1, 1)$ and $\beta \in \mathcal{B} \sim \mathcal{U}(-1, 1)$; Out of distribution: we used $\{(\alpha, \beta) \in (\mathcal{A}, \mathcal{B}) \sim (\mathcal{U}(-1, 1.2)), \mathcal{U}(1, 1.2)) \cup (\mathcal{U}(1, 1.2)), \mathcal{U}(-1.2, 1)) \cup (\mathcal{U}(-1.2, 1)), \mathcal{U}(-1.2, -1)) \cup (\mathcal{U}(-1.2, -1)), \mathcal{U}(-1, 1.2))\}$.

## E  Policy Design

We show the detailed policy design in this section. All the baseline policies were trained to minimize the trajectory reproduction error.
**KODex:** The representation of the system is given as: $x_r = [x_r^1, x_r^2, \cdots, x_r^n]$ and $x_o = [x_o^1, x_o^2, \cdots, x_o^m]$ and superscript is used to index states. The details of the state design for each task is provided in Appendix B. In experiments, the vector-valued lifting functions $\psi_r$ and $\psi_o$ in (8)

were polynomial basis function defined as

$$\psi_r = \{x_r^i x_r^j\} \cup \{(x_r^i)^3\} \text{ for } i, j = 1, \cdots, n$$
$$\psi_o = \{x_o^i x_o^j\} \cup \{(x_o^i)^2 (x_o^j)\} \text{ for } i, j = 1, \cdots, m \tag{11}$$

Note that $x_r^i x_r^j / x_r^j x_r^i$ only appears once in lifting functions (similar to $x_o^i x_o^j / x_o^j x_o^i$), and we ignore $t$ as the lifting functions are the same across the time horizon.

The choice of lifting functions can be viewed as the hyper-parameter of KODex. We make this choice as inspired from [25] and experimental results also indicate its effectiveness. Through all the experiments, we sticked with the same set of lifting functions, which helped to relieve us from extensive efforts of tuning the hyper-parameters, e.g. network layer size, that were necessary for baseline policies as shown in Appendix F.

**Full-connected Neural Network (NN):** The first baseline is a feedforward network that ingests the states $x(1)$ and iteratively produces the predictions $x(t), t = 2, \cdots, T$ via the rollout of a Multilayer Perceptron (MLP). The reference joint trajectories $(x_r(t))$ are then used to execute the robot with the learned controller $C$. The significance of this baseline is to evaluate a policy that produces a high-dimensional motion without any additional structure.

**Long Short-Term Memory (LSTM):** We create an LSTM-based policy under the same input-output flow as the NN policy. We also apply two fully-connected layers between the task input/output and the input/hidden state of the LSTM network. Similarly, the same controller $C$ is deployed to track the reference joint trajectory. LSTM networks are known to be beneficial to imitation learning [33] and suitable for sequential processing [42], e.g, motion generation. Therefore, we expect to evaluate the performance of the recurrent structures in these tasks.

**Neural Dynamic Policy (NDP):** The Neural Dynamic Policy [18] embeds desired dynamical structure as a layer in neural networks. Specifically, the parameters of the second order Dynamics Motion Primitives (DMP) are predicted as outputs of the preceding layers (MLP in [18]). As a result, it allows the overall policy easily reason in the space of trajectories and can be utilized for learning from demonstration. We train an NDP policy following the imitation learning pipeline described in [18]. For each task, given $x(1)$, the neural network components in NDP generate the parameters of DMPs (radial basis functions (RBFs) in [18]), which are forward integrated to produce the reference joint trajectories for tracking.

**Neural Geometric Fabrics policy (NGF):** The Neural Geometric Fabrics [4], a structured policy class, that enables efficient skill learning for dexterous manipulation from demonstrations by leveraging structures induced by Geometric Fabrics [43]. Geometric Fabrics is a stable class of the Riemannian Motion Policy (RMP) [44]. It has been demonstrated that NGF outperforms RMP in policy learning for dexterous manipulation task in [4]. The NGF policy is defined in the configuration space of the robot, which is composed of a geometric policy, a potential policy and a damping term. More specifically, the NGF policy is constructed as follows: (1) define a geometric policy pair $[\mathbf{M}, \pi]$ and a potential policy pair $[\mathbf{M}_f, \pi_f]$ in the configuration space $\mathbf{q}$, (2) energize the geometric policy (project orthogonal to the direction of motion with $\mathbf{p}_e$) to create a collection of energy-preserving paths (the Geometric Fabric), and (3) force the Geometric Fabric with a potential defined by $[\mathbf{M}_f, \pi_f]$ and damp via $b$ applied along $\dot{\mathbf{q}}$, which ensures convergence to the potential's minima. The potential policy $\pi_f$ is the gradient of a function of position only. Note that we parameterize the geometric policy pair $[\mathbf{M}, \pi]$, the potential policy pair $[\mathbf{M}_f, \pi_f]$, and the damping scalar $b$ with MLP networks and learn them from demonstration data.

## F   Optimizing baseline model size

As described in Appendix E, we sticked with the same set of lifting functions for KODex and report the task success rate when we trained KODex on training set and tested it on validation set in Table. 1. However, for baselines, the hyper-parameters were selected through a set of ablation experiments for each task using the training set over three choices of model size, including small size, median size and large size. We generated five random seeds for parameter initialization per model size, per baseline, and per task, as all learning based baseline models are sensitive to parameter initialization [24]. For each baseline policy, we report the mean and standard deviation of the task success rate on the validation set over five random seeds in Tables. 2-5.

Based on these results, we selectd the model size that offers the best performance in terms of task success rate. In addition, these results indicate that, unlike KODex, extensive hyper-parameter tun-

ing and various trials on parameter initialization for baseline models are necessary. Note that we use $l$ to denote $\dim(\mathrm{x}(t))$.

Table 1: Task success rate on validation set (KODex)

| Tool | Door | Relocation | Reorientation |
|------|------|------------|---------------|
| 100.0% | 96.0% | 88.0% | 62.0% |

Table 2: Hyper-parameters on NN Network Sizes

| Success Rate (%) Model Size — Task | Tool | Door | Relocation | Reorientation |
|---|---|---|---|---|
| MLP: (32, 64, 32) | **0.4($\pm$0.8)** | 0.0($\pm$0.0) | 0.4($\pm$0.8) | 6.8($\pm$3.9) |
| MLP: (64, 128, 64) | 0.0($\pm$0.0) | **0.4($\pm$0.8)** | **1.2($\pm$2.4)** | **10.4($\pm$6.6)** |
| MLP: (128, 256, 128) | 0.0($\pm$0.0) | 0.0($\pm$0.0) | 0.8($\pm$1.6) | 6.0($\pm$1.5) |

Table 3: Hyper-parameters on LSTM Network Sizes

| Success Rate (%) Model Size — Task | Tool | Door | Relocation | Reorientation |
|---|---|---|---|---|
| LSTM: 200 fc: ($l$, 100), (200, $l$) | 28.8($\pm$25.0) | **87.6($\pm$10.3)** | 7.6($\pm$5.9) | **56.4($\pm$7.4)** |
| LSTM: 250 fc: ($l$, 175), (250, $l$) | **60.8($\pm$36.6)** | 80.8($\pm$24.5) | 7.6($\pm$7.5) | 48.0($\pm$17.0) |
| LSTM: 300 fc: ($l$, 250), (300, $l$) | 44.8($\pm$31.8) | 82.0($\pm$13.9) | **16.4($\pm$14.5)** | 54.0($\pm$11.0) |

Table 4: Hyper-parameters on NDP Network Sizes

| Success Rate (%) Model Size — Task | Tool | Door | Relocation | Reorientation |
|---|---|---|---|---|
| MLP: (32, 64, 32) 10 RBFs | 0.0($\pm$0.0) | 8.0($\pm$2.5) | 30.0($\pm$9.3) | 57.2($\pm$8.6) |
| MLP: (64, 128, 64) 20 RBFs | 16.8($\pm$29.8) | 40.8($\pm$8.1) | 74.0($\pm$4.9) | 59.2($\pm$6.5) |
| MLP: (128, 256, 128) 30 RBFs | **18.4($\pm$31.9)** | **66.0($\pm$5.2)** | **79.2($\pm$7.7)** | **62.4($\pm$7.8)** |

Table 5: Hyper-parameters on NGF Network Sizes

| Success Rate (%) Model Size — Task | Tool | Door | Relocation | Reorientation |
|---|---|---|---|---|
| MLP: (64, 32) | 99.2($\pm$1.6) | 87.2($\pm$12.0) | 87.6($\pm$8.5) | 77.6($\pm$2.3) |
| MLP: (128, 64) | **100.0($\pm$0.0)** | 90.0($\pm$5.9) | 94.4($\pm$3.2) | 72.4($\pm$4.5) |
| MLP: (256, 128) | 83.6($\pm$20.1) | **90.8($\pm$4.3)** | **95.2($\pm$1.6)** | **78.4($\pm$3.4)** |

# G   Hyper-parameters for controller learning

The hyper-parameters we used to learn the inverse dynamics controller $C$ for each task were the same as listed in Table. 6. Note that we use $l_r$ to denote $\dim(\mathrm{x}_r(t))$.

Table 6: Hyper-parameters on controller learning

| Hidden Layer | Activation | Learning Rate | Iteration |
|---|---|---|---|
| $(4l_r, 4l_r, 2l_r)$ | ReLU | 0.0001 | 300 |

Table 7: One-step inference time (in milliseconds) with mean and standard deviation over the task horizon.

| Policy \ Task | Tool | Door | Relocation | Reorientation |
|---|---|---|---|---|
| NN | 1.39($\pm$0.26) | 1.26($\pm$0.39) | 1.02($\pm$0.09) | 1.15($\pm$0.12) |
| LSTM | 1.71($\pm$0.28) | 1.32($\pm$0.34) | 1.59($\pm$0.57) | 1.42($\pm$0.13) |
| NDP | 1.88($\pm$0.30) | 1.08($\pm$0.22) | 1.05($\pm$0.06) | 1.32($\pm$0.21) |
| NGF | 1.37($\pm$0.16) | 1.17($\pm$0.26) | 1.72($\pm$0.36) | 1.19($\pm$0.12) |
| KODex | 1.71($\pm$0.48) | 1.04($\pm$0.27) | 1.12($\pm$0.24) | 1.08($\pm$0.60) |

## H  Inference Time

We report the inference time for each method in Table. 7. Our results indicate that KODex's inference time is on par with the SOTA baselines. As such, it reveals that KODex can be translated to physical hardware and meet necessary control frequency.

## I  Zero-Shot Out-of-Distribution Generalization

We generated a new set of 10,000 out-of-distribution samples to evaluate how the policies that were trained on 200 demonstrations generalize to unseen samples (see Appendix D for details on the sampling procedure). In Fig. 5, we report the task success rates of each method trained on the 200 demonstrations and tested on the 10,000 out-of-distribution samples. In addition, we also report the task success rate of the expert policy on the same 10,000 out-of-distribution samples to establish a baseline. Perhaps unsurprisingly, none of the methods are able to consistently outperform the expert policy in most tasks. We observe that KODex is able to outperform the four baselines in Tool Use task. In the other tasks, the highly-structured NGF performs the best, and KODex's performs comparably to NDP and LSTM.

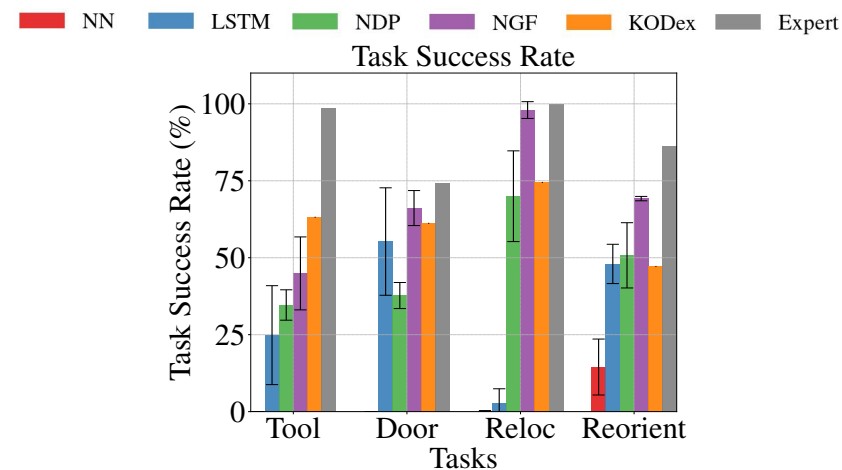

Figure 5: Zero-Shot Out-of-distribution task success rates

## J  Robustness to changes in physical properties

We evaluate the robustness of the reference dynamics learned by each method to changes in hand mass or object mass for each task. This experiment is motivated by the fact that sim-to-real transfer often involves changes in physical properties. Further, consistent use of robotic hardware could result in changes to physical properties. Specifically, we consider four variations per task:

- Tool Use: i) *Heavy Object (Hammer)*: 0.25 (default) → 0.85 (new), ii) *Light Object (Hammer)*: 0.25 (default) → 0.10 (new), iii) *Light Hand (Palm)*: 4.0 (default) → 1.0 (new), and iv) *Heavy Hand (Palm)*: 4.0 (default) → 8.0 (new)
- Door: i) *Heavy Object (Latch)*: 3.54 (default) → 12.54 (new), ii) *Light Object (Latch)*: 3.54 (default) → 0.54 (new), iii) *Light Hand (Palm)*: 4.0 (default) → 1.5 (new), and iv) *Heavy Hand (Palm)*: 4.0 (default) → 7.0 (new)
- Relocation: i) *Heavy Object (Ball)*: 0.18 (default) → 1.88 (new), ii) *Light Object (Ball)*: 0.18 (default) → 0.05 (new), iii) *Light Hand (Palm)*: 4.0 (default) → 3.0 (new), and iv) *Heavy Hand (Palm)*: 4.0 (default) → 5.0 (new);
- Reorientation: i) *Heavy Object (Pen)*: 1.5 (default) → 9.5 (new), ii) *Light Object (Pen)*: 1.5 (default) → 0.2 (new), iii) *Light Hand (Finger Knuckles)*: 0.008 (default) → 0.0001 (new), and iv) *Heavy Hand (Finger Knuckles)*: 0.008 (default) → 0.20 (new)

It is important to note we held the reference dynamics learned by each method constant for this experiment, irrespective of the changes to the hand or the object. Instead, we relearned the tracking controller using 200 rollouts from the expert agent, following the procedure detailed in Section. 4.3.

In Tables. 8-11, we report the task success rate of KODex, and other baseline policies (all trained on 200 demonstrations) before and after relearning the controller. We also report the task success rates of the expert agents to establish baselines.

We find that the Light Hand variation results in the lowest drop in performance across all methods and all tasks, thus consequently relearning controllers does not offer any considerable improvements. In contrast, all methods benefit from relearning the controller in the Heavy Hand variations, as evidenced by the increased task success rates. Overall, we find that KODex outperforms all baselines, with the exception of NGF which performs better than KODex under a few variations and tasks. Surprisingly, KODex (and some baselines) when used with the original controller outperform the expert policy under a few variations (e.g., Heavy Object in Relocation task, and Heavy Object in Door task). We believe this is due to the fact that KODex and the baselines learn to generate and track desired trajectories separately, while the expert RL directly generates control inputs from state information. In particular, the learned desired trajectories for a given tasks are likely invariant to slight changes in physical properties. On rare occasions where this is not the case, we indeed find that fine-tuning the tracking controllers worsens the performance.

These results demonstrate that changes to the robot/system dynamics can be handled by fine tuning the tracking controller without the need for relearning the reference dynamics. Once again, KODex is able to perform comparably to or outperform SOTA approaches despite its simplicity.

Table 8: Robustness to variations in the physical properties (Tool Use)

| Success Rate (%) / Variation / Controller | Heavy Object | Light Object | Light Hand | Heavy Hand |
|---|---|---|---|---|
| Expert agent | 93.5 | 66.2 | 65.4 | 71.2 |
| KODex + Original controller | 46.0 | 64.0 | 99.5 | 46.5 |
| NN + Original controller | 0.0($\pm$0.0) | 0.0($\pm$0.0) | 0.0($\pm$0.0) | 0.7($\pm$1.4) |
| LSTM + Original controller | 32.7($\pm$18.7) | 35.0($\pm$22.1) | 44.3($\pm$23.1) | 52.7($\pm$27.5) |
| NDP + Original controller | 0.0($\pm$0.0) | 68.0($\pm$20.8) | 45.4($\pm$37.4) | 0.0($\pm$0.0) |
| NGF + Original controller | 33.4($\pm$11.5) | 62.9($\pm$27.5) | 83.2($\pm$26.3) | 40.3($\pm$20.2) |
| KODex + Expert-tuned controller | 53.5 | 44.0 | 89.0 | 92.5 |
| NN + Expert-tuned controller | 0.0($\pm$0.0) | 0.0($\pm$0.0) | 0.2($\pm$0.4) | 0.0($\pm$0.0) |
| LSTM + Expert-tuned controller | 42.4($\pm$34.3) | 33.7($\pm$14.9) | 52.2($\pm$22.7) | 69.9($\pm$19.4) |
| NDP + Expert-tuned controller | 33.3($\pm$20.0) | 23.8($\pm$24.4) | 29.4($\pm$37.1) | 39.8($\pm$24.5) |
| NGF + Expert-tuned controller | 48.2($\pm$18.0) | 48.7($\pm$12.2) | 94.6($\pm$8.9) | 82.1($\pm$7.5) |

## K    The impact of the choice of basis functions

We evaluate if KODex's performance is impacted by different sets of polynomial functions that are used as the lifting function. We trained all policies on 200 demos and tested them on 10,000 unseen initial conditions.

Table 9: Robustness to variations in the physical properties (Door)

| Success Rate (%)  Variation  Controller | Heavy Object | Light Object | Light Hand | Heavy Hand |
|---|---|---|---|---|
| Expert agent | 45.2 | 91.7 | 82.0 | 74.9 |
| KODex + Original controller | 57.0 | 97.0 | 56.5 | 33.5 |
| NN + Original controller | 0.0($\pm$0.0) | 0.2($\pm$0.4) | 1.3($\pm$2.1) | 0.0($\pm$0.0) |
| LSTM + Original controller | 34.4($\pm$8.7) | 75.8($\pm$19.5) | 38.1($\pm$10.4) | 33.5($\pm$11.4) |
| NDP + Original controller | 22.1($\pm$1.9) | 62.8($\pm$5.2) | 51.1($\pm$4.9) | 3.1($\pm$2.3) |
| NGF + Original controller | 48.7($\pm$6.7) | 95.0($\pm$2.1) | 42.1($\pm$11.0) | 33.8($\pm$10.0) |
| KODex + Expert-tuned controller | 39.0 | 94.0 | 54.0 | 81.5 |
| NN + Expert-tuned controller | 0.0($\pm$0.0) | 0.0($\pm$0.0) | 0.7($\pm$0.9) | 0.0($\pm$0.0) |
| LSTM + Expert-tuned controller | 21.2($\pm$5.3) | 75.4($\pm$18.0) | 49.2($\pm$8.1) | 56.9($\pm$18.7) |
| NDP + Expert-tuned controller | 15.5($\pm$3.0) | 36.2($\pm$10.6) | 25.5($\pm$4.4) | 8.8($\pm$3.0) |
| NGF + Expert-tuned controller | 36.6($\pm$5.1) | 95.5($\pm$1.8) | 57.7($\pm$4.7) | 77.1($\pm$6.7) |

Table 10: Robustness to variations in the physical properties (Relocation)

| Success Rate (%)  Variation  Controller | Heavy Object | Light Object | Light Hand | Heavy Hand |
|---|---|---|---|---|
| Expert agent | 77.0 | 100.0 | 100.0 | 100.0 |
| KODex + Original controller | 19.5 | 89.5 | 82.5 | 21.5 |
| NN + Original controller | 0.1($\pm$0.2) | 1.6($\pm$2.5) | 1.5($\pm$2.1) | 1.7($\pm$2.2) |
| LSTM + Original controller | 0.4($\pm$0.4) | 15.4($\pm$10.7) | 9.5($\pm$8.1) | 7.7($\pm$9.4) |
| NDP + Original controller | 13.5($\pm$5.0) | 85.6($\pm$8.1) | 72.1($\pm$9.6) | 31.6($\pm$10.0) |
| NGF + Original controller | 25.8($\pm$4.9) | 96.4($\pm$1.4) | 96.6($\pm$0.97) | 19.3($\pm$3.8) |
| KODex + Expert-tuned controller | 34.0 | 93.0 | 85.0 | 89.0 |
| NN + Expert-tuned controller | 0.2($\pm$0.4) | 0.6($\pm$0.7) | 1.4($\pm$1.8) | 1.5($\pm$2.3) |
| LSTM + Expert-tuned controller | 5.8($\pm$4.7) | 15.2($\pm$12.5) | 15.5($\pm$10.7) | 14.1($\pm$9.3) |
| NDP + Expert-tuned controller | 19.9($\pm$5.8) | 84.5($\pm$8.9) | 63.2($\pm$15.0) | 92.4($\pm$1.2) |
| NGF + Expert-tuned controller | 52.6($\pm$3.6) | 98.1($\pm$1.2) | 95.6($\pm$2.2) | 94.5($\pm$0.9) |

Table 11: Robustness to variations in the physical properties (Reorientation)

| Success Rate (%)  Variation  Controller | Heavy Object | Light Object | Light Hand | Heavy Hand |
|---|---|---|---|---|
| Expert agent | 46.8 | 69.0 | 95.2 | 89.7 |
| KODex + Original controller | 53.5 | 55.0 | 66.5 | 61.5 |
| NN + Original controller | 4.7($\pm$2.6) | 9.6($\pm$8.1) | 9.5($\pm$6.4) | 7.9($\pm$6.5) |
| LSTM + Original controller | 34.5($\pm$7.8) | 52.3($\pm$10.6) | 60.3($\pm$6.0) | 55.6($\pm$7.8) |
| NDP + Original controller | 49.4($\pm$3.6) | 58.4($\pm$6.4) | 59.8($\pm$7.6) | 55.7($\pm$9.7) |
| NGF + Original controller | 39.9($\pm$1.9) | 57.1($\pm$2.2) | 81.6($\pm$1.8) | 73.4($\pm$3.8) |
| KODex + Expert-tuned controller | 52.0 | 63.0 | 71.5 | 65.5 |
| NN + Expert-tuned controller | 1.5($\pm$0.9) | 5.2($\pm$4.2) | 3.8($\pm$1.7) | 3.7($\pm$2.6) |
| LSTM + Expert-tuned controller | 43.5($\pm$7.9) | 47.7($\pm$8.8) | 61.4($\pm$4.2) | 54.4($\pm$5.5) |
| NDP + Expert-tuned controller | 55.5($\pm$5.9) | 59.0($\pm$5.5) | 63.0($\pm$6.5) | 57.0($\pm$7.5) |
| NGF + Expert-tuned controller | 49.1($\pm$2.6) | 59.7($\pm$3.2) | 79.4($\pm$1.9) | 72.6($\pm$1.2) |

**Design**: Specifically, we define four sets of observables (one of which was used in the original submission). Let robot state: $x_r = [x_r^1, x_r^2, \cdots, x_r^n]$ and $x_o = [x_o^1, x_o^2, \cdots, x_o^m]$ denote the robot and the object state, respectively, with superscript indexing the states. We then define four vector-valued lifting functions $\psi_r$ and $\psi_o$ in (8) as follows

- Set 1

$$\psi_r = \{(x_r^i)^2\} \text{ for } i = 1, \cdots, n$$
$$\psi_o = \{(x_o^i)^2\} \text{ for } i = 1, \cdots, m$$

- Set 2

$$\psi_r = \{x_r^i x_r^j\} \text{ for } i,j = 1, \cdots, n$$
$$\psi_o = \{x_o^i x_o^j\} \text{ for } i,j = 1, \cdots, m$$

- **Set 3 (used in this work)**

$$\psi_r = \{x_r^i x_r^j\} \cup \{(x_r^i)^3\} \text{ for } i,j = 1, \cdots, n$$
$$\psi_o = \{x_o^i x_o^j\} \cup \{(x_o^i)^2 (x_o^j)\} \text{ for } i,j = 1, \cdots, m$$

- Set 4

$$\psi_r = \{x_r^i x_r^j\} \cup \{(x_r^i)^2 (x_r^j)\} \text{ for } i,j = 1, \cdots, n$$
$$\psi_o = \{x_o^i x_o^j\} \cup \{(x_o^i)^2 (x_o^j)\} \text{ for } i,j = 1, \cdots, m$$

We report the number of observables for each set and task combination in Table. 12.

Table 12: Number of observables

| Set \ Task | Tool n=26,m=15 | Door n=28,m=7 | Relocation n=30,m=12 | Reorientation n=24,m=12 |
|---|---|---|---|---|
| Set 1 | 82 | 70 | 84 | 72 |
| Set 2 | 512 | 469 | 585 | 414 |
| **Set 3** (ours) | 763 | 546 | 759 | 582 |
| Set 4 | 1413 | 1302 | 1629 | 1134 |

Figure 6: The effects of lifting function on training time (left), imitation error (center), and success rate (right).

**Discussion**: As shown in Fig. 6, it is clear that training time increases with the number of observables since the Moore–Penrose inverse requires more computation for higher-dimension matrices. Importantly, KODex's success rate across all tasks remained roughly the same for Sets 2, 3, and 4. In general, as one would expect, increasing the number of observables tends to decrease imitation error and increase task success rate. The only exception to this trend is observed for the Object Relocation task, in which KODex performs marginally better when trained on Set 2 (585 observables) compared with it trained on Set 3 (759 observables). Taken together, these results suggest that KODex's performance is not highly sensitive to the specific choice of lifting function, as long as sufficient expressivity is ensured.

## L  Stability Analysis

Another unique advantage of utilizing Koopman Operators to model the reference dynamics for dexterous manipulation tasks is that the learned policy is a linear dynamical system which can be readily inspected and analyzed, in stark contrast to SOTA methods built upon deep neural networks.

We analyzed the stability of the learned policy. For a linear dynamical system with complex conjugate eigenvalues $\lambda_i = \theta_i \pm j\omega_i$, i.e., KODex with Koopman matrix $\mathbf{K}$, the system is asymptotically stable if all of the eigenvalues have magnitude ($\rho_i = \sqrt{\theta_i^2 + \omega_i^2}$) less than one. From the standpoint

of control theory, it is beneficial to have a asymptotically stable system because of the guarantee that all system states will converge. However, from the standpoint of dexterous manipulation tasks considered in this work, strict stability might not be preferable because the final desired hand poses and object poses are not identical for different initial conditions. This represents a natural trade-off between safety and expressivity. As such, understanding how KODex addresses this trade-off can be illuminating.

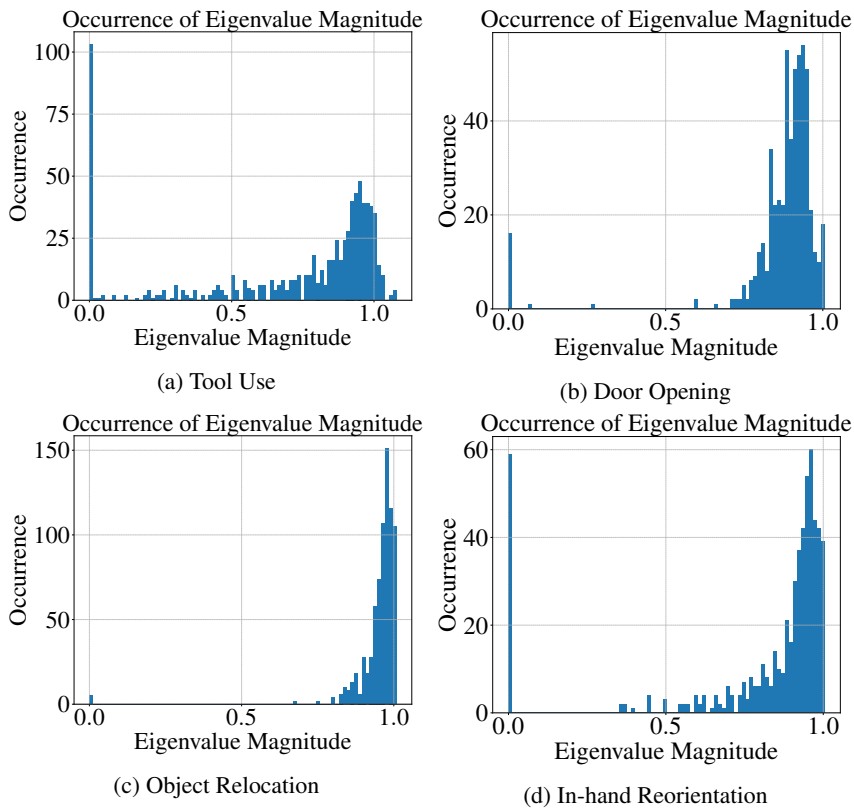

Figure 7: Occurrence of Eigenvalue Magnitude

Table 13: Maximun Eigenvalue Magnitude

| Tool Use | Door Opening | Object Relocation | In-hand Reorientation |
|----------|--------------|-------------------|-----------------------|
| 1.07888  | 1.00553      | 1.00859           | 1.00413               |

In Fig. 7, we report a histogram of the Koopman matrix's eigenvalue magnitudes in each task. In addition, we report the maximum eigenvalue magnitude in Table. 13. Based on these results, we can see that i) most eigenvalues' magnitudes are less than one, suggesting that KODex tends to learn nearly-stable policies that generate *safe trajectories* during execution, and ii) a few eigenvalues have magnitude larger than one, suggesting KODex does not prioritize stability, at the expense of expressivity required to achieve the reported performance.

## M   Comparisons against Behaviour Cloning with State-Action Mapping

We conducted an additional experiment involving a new neural network based baseline policy that learns to directly map states $x(t)$ to actions $\tau(t)$ instead of learning the reference dynamics and the tracking controller. The new policy (State-action BC) was built upon the MLP architecture with three hidden layers ([64, 128, 64]), and was trained over three random seeds to minimize the state-action reproduction error. For a fair comparison, these policies were trained and tested on the same set of demonstrations and testing instances as in Section. 5.3.

In Fig. 8, we report the training time and the task success rate on each task for KODex and State-

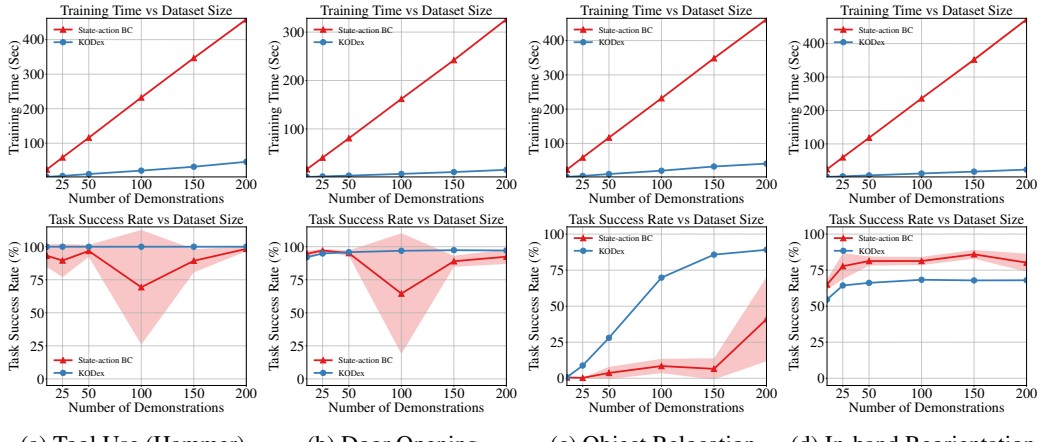

(a) Tool Use (Hammer)  (b) Door Opening  (c) Object Relocation  (d) In-hand Reorientation

Figure 8: The effects of number of demonstrations on training time (top row), and success rate (bottom row) for KODex and State-action BC on each task. Solid lines indicate mean trends and shaded areas show ± standard deviation over three random seeds.

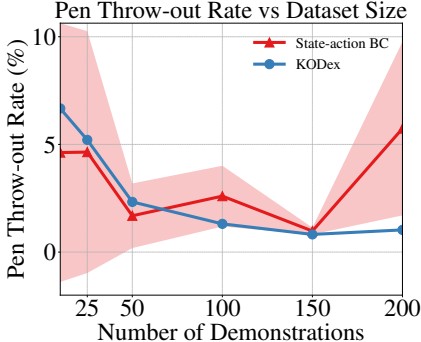

Figure 9: The effects of number of demonstrations on pen throw-out rate for KODex and State-action BC.

Action BC. The results reveal a familiar trend: across all tasks, KODex is drastically more computationally efficient than State-Action BC, while performing comparably (if not better) in terms of success rate.

Further, we would like to highlight two other advantages of KODex over State-Action BC. First, KODex could be potentially applied on state-only demonstrations, with a manually-tuned PD controller replacing the learned tracking controller (one could also learn the controller via reinforcement learning [39]). In contrast, state-action imitation learning methods inevitably need action labels. Second, KODex is safer for online execution. Since KODex separates motion generation and tracking, it tends to take less risky actions. But state-action policies may take unsafe actions when they encounter unseen states due to covariate shift. In Fig. 9, we report the pen throw-out rate from the In-hand Reorientation task. It can be seen that the State-action BC policy is more likely to generate undesirable behaviours, resulting in complete task failures. This implies that KODex may be safer for hardware implementations, thanks to the separation of reference motion and tracking. Although there are a few other state-action policies that better address covariate shift (e.g., GAIL [45]), such comparisons are outside of scope of this work.

