# OpenReview forum: "On the Utility of Koopman Operator Theory in Learning Dexterous Manipulation Skills"
_robot-learning.org/CoRL/2023/Conference — CoRL 2023 Oral_

### Official Review · Reviewer_k76a · 2023-07-20

**Confidence:** 4
**Originality:** Good
**Technical Quality:** Very Good
**Clarity Of Presentation:** Very Good
**Impact:** 3

**Recommendation:**

Weak Accept: I recommend accepting the paper, but will not argue for my recommendation if the majority of other reviewers have a different opinion.

**Review:**

Overall the idea of using koopman operators for modeling linear dynamics is reasonable, but is not new in MBRL, as the many citations in this paper also represent. I think it’s still worthwhile to understand how well this can work in the context of dexterous manipulation though.

The part I find most confusing about this paper in the way it is written right now is - where is the Koopman operator K actually used? If it’s simply an imitation learning framework, why do we need the Koopman operator at all? There seems to be a missing section on how the K is used to generate the reference trajectory - via LQR or something else? If it’s imitation learning and the trajectories are given to you then why not just directly track those? And if it’s about generating reference trajectories in new initial states, then is it just rolling out the Koopman operator forward since the actions are implicit in the dynamics? I suspect it is this, but this should be made very clear in the paper.

The basic premise of this work seems to be the fact that learning Koopman operators give you an easy way to roll out the trajectories in new scenarios, and these can be tracked with inverse dynamics. I think this should be compared with standard BC / IRL type methods on this task, which may work quite well.

The claim is made that the other techniques need things like gradient descent etc, which is true for NN policies, but there are other structured policies that could be used like differentiable MPC - or like guided policy search which can be extremely data efficient. Those might be more reasonable baselines to compare to.

Also the choice of \phi seems very important or at least I’d expect it to be. Can the authors provide intuition on why this doesn’t matter?

“While counter-intuitive, this 241 observation follows the recent finding that imitation error and task success rate might not necessarily 242 be correlated” —> can more be said about this?

I think the demonstration is reasonable - but given the somewhat limited novelty in directly learning koopman dynamics and using it for imitation learning, a deeper empirical analysis - more comparisons, more domains (eg from PDDM (nagabandi et al)), and understanding the failure modes more clearly would be very helpful.


Strengths:
1. Simple method, seems like it has strong empirical performance.
2. Big improvement in sample efficiency and in robustness which is powerful for imitation learning settings.

Weaknesses:
1. More comparisons would be helpful, especially to techniques like guided policy search or differentiable MPC.
2. A deeper understanding of failure modes would be helpful in understanding the tradeoffs of using Koopman versus other policy classes for imitation.
3. An analysis across some more domains (eg PDDM) would help add to the paper.
4. A comparison with standard imitation methods like IRL/BC would also be helpful.


**Quality Of The Limitations Section:**

Limitations are addressed clearly

**Questions For Rebuttal:**

Repeating weaknesses here:
1. More comparisons would be helpful, especially to techniques like guided policy search or differentiable MPC.
2. A deeper understanding of failure modes would be helpful in understanding the tradeoffs of using Koopman versus other policy classes for imitation.
3. An analysis across some more domains (eg PDDM) would help add to the paper.
4. A comparison with standard imitation methods like IRL/BC would also be helpful.

**Robotics Focus:**

Highly relevant to robotics but no hardware experiments

**Summary Of Paper:**

This paper proposes a technique for modeling the dynamics of dexterous manipulation problems as linear dynamics in a lifted high dimensional space, using koopman operator theory. They then show the ability to learn this koopman operator from data as a means to represent linear dynamics in this lifted space. They then use this learned linear dynamics to generate reference trajectories and learn an inverse dynamics model with an MLP to track the generated reference trajectories.


**Summary Of Recommendation:**

Overall i like the simplicity and effectiveness of the approach, but I have suggested some additional empiricism that may help strengthen the paper. If these points are addressed, I'm willing to raise my score.

---

### Official Review · Reviewer_RjjK · 2023-07-21

**Confidence:** 3
**Originality:** Good
**Technical Quality:** Good
**Clarity Of Presentation:** Very Good
**Impact:** 3

**Recommendation:**

Weak Accept: I recommend accepting the paper, but will not argue for my recommendation if the majority of other reviewers have a different opinion.

**Review:**

Quality: The overall quality of the work is good as it addresses an important challenge in using learning-based approaches, by adopting theories from control such as the Koopman operator theory.

Clarity: The aims, methods and experiments are clear, while the appendix is also very detailed.

Originality: The originality of the work is good, although there has been several efforts towards combining control theories with learning-based methods for improving sample efficiency and scalability.

Significance of the work: Overall, the significance could be further improved if the experiments were deployed and the choice of task environments further motivated.

Main Strengths: The main strengths of the paper are the clarity and logical way the paper is presented, the fact that they clearly outline what KODex aims to achieve in comparison to other related works and the experimental evaluation is very comprehensive.

Weaknesses: The main weaknesses of the paper are the fact that the experiments were carried out in simulation and I am not sure these would be easily deployed in the real world (although this is addressed in the limitations section).

**Quality Of The Limitations Section:**

Limitations are addressed clearly

**Questions For Rebuttal:**

(1) "First, implementing existing algorithms requires significant technical expertise and modern machine learning infrastructure." -> could you elaborate more on "modern machine learning infrastructure"? This seems a bit vague.

(2) "Third, existing approaches demand painstaking effort in tuning hyperparameters to achieve acceptable performance". What do you really mean by "acceptable performance" as there have been methods that have achieve SOTA performance and not only acceptable? This sentence seems to downplay the strengths of learning-based methods.

(3) In Section 4.3: "a PID controller could be used instead of learning a tracking controller."  - Why do the authors recommend a PID and not a PD controller as is common in manipulation tasks?

(4) Following from (3), a major challenge in learning controllers is the inability to guarantee easily stability. Did the authors think about this for their methods and how do they ensure their controller remains stable?

(5) Also in Section 4.3: "one would have to painstakingly tune the control gains and frequency, and do so for each task independently" - Would the authors not argue it's the same for learning-based methods with the hyperparameter optimisation?

(6) The authors use training time (time taken to train a policy) as an evaluation metric - why do they use only the training time and not inference time?

(7) I would like to know the authors' thoughts on the choice of the 4 tasks - why these particular 4 (are they challenging their method in different aspects?). Also, did they consider how their method would generalise from task to task (e.g. learn in the easiest task and then transfer the policy to harder tasks?)

**Robotics Focus:**

Relevant but unlikely to deploy to hardware in near future

**Summary Of Paper:**

The authors investigate the utility of Koopman operator theory in alleviating the limitations of existing learning-based approaches, with a particular focus on general efficacy, computational efficiency, sample efficiency, and scalability. They develop a novel imitation learning framework, dubbed Koopman Operator-based Dexterous Manipulation (KODex), to evaluate the utility of Koopman operator theory for dexterous manipulation. Their experimental evaluation is carried out on four dexterous manipulation skills on the simulated Adroit hand: Tool Use, Door Opening, Relocation, and Reorientation. They benchmarked their method against NN and LSTM (unstructured baselines) and NDP and NGF (highly-structured baselines). They conclude that their KODex can i) analytically learn dexterous manipulation skills and ii) match or outperform SOTA imitation learning approaches on various dexterous manipulation tasks, while being an order of magnitude faster.


**Summary Of Recommendation:**

I would recommend this paper to be accepted, following some further thoughts/maybe preliminary experiments on using robot hardware and addressing the generalisation of the methods across different tasks (e.g. training on a task, testing on another similar task).

---

### Official Review · Reviewer_aS5H · 2023-08-02

**Confidence:** 3
**Originality:** Good
**Technical Quality:** Very Good
**Clarity Of Presentation:** Excellent
**Impact:** 4

**Recommendation:**

Strong Accept: I recommend accepting the paper and will argue for my recommendation even if other reviewers hold a different opinion.

**Review:**

Overall, a solid paper with meaningful contributions to the field of robotics and manipulation. The paper is well written, with a solid background and introduction, a clear and concise description of the method, and meaningful examples with easy-to-interpret results.

The results of the paper are noteworthy: they achieve performance on par with recent state-of-the-art methods with a method that takes significantly less training time. While there are many areas for improvement and future work, the community will benefit from the results of this paper, which clearly show that learning methods outside the realm of neural networks and deep reinforcement learning have exceptional merit and are worth exploring.

Strengths:
* Clearly written
* Impressive results
* Use of a lesser-known learning technique
* Clear and honest evaluation of current learning-based methods for manipulation
* Consideration and evaluation of training time as an important metric

Weaknesses:
* Limitations and future work section is lacking
* Description of the overall training pipeline is somewhat unclear
* Confusion over what dynamics are being learned by the method
* A few areas where the theory is glossed over or not expressed correctly
* Too many discussion points are deferred to Appendices
* Lack of comparison with non-learning approaches
* Lack of demonstration of physical hardware (minor, given it's noted in the limitations section and left for future work)

**Quality Of The Limitations Section:**

Additional details required

**Questions For Rebuttal:**

# Primary Concerns
1. Is the method actually learning the underlying dynamics of the system? My principal concern is about the use of "open-loop" reference trajectory generation. After learning the Koopman matrix from expert demonstrations, the reference trajectory is generated by simulating the linear system forward one time step. The assumption here is that the "learned dynamics" somehow encode the task. The true underlying dynamics (based on physics) should be task-agnostic, so the authors aren't truly trying to learning the underlying dynamics of the system, but rather the *dynamics of the task.* I think this is an important distinction that should be made. I'm not saying this is a bad idea, just that the authors should be more careful in making this distinction. How could this method be used to actually the learn underlying system dynamics and improve task generalization?
2. Why did the authors not compare against classical model-based methods? The authors state several times that one of the advantages of Koopman-based methods it that it allows the use of linear-systems and control-theoretic approaches, but other than a stability analysis included in the Appendix, this isn't really leveraged.
3. Did the authors try to use an LQR tracking controller instead of a learned one?
4. In Eq 8, you don't include any non-linear coupling of the robot and the object. Why?
5. Why did the authors choose polynomials? The types of polynomials chosen by the authors are usually poorly conditioned. Better choices include options like Chebyshev polynomials. Also, manipulation tasks include friction and contact, which are both non-smooth, so why not include non-smooth functions like ReLU?

# Secondary Questions
6. In the third paragraph of 4.1, the authors state the system dynamics are "arbitrarily complex." What do you mean by this? What is arbitrarily complex? We have well-understood models of rigid-body dynamics, and decent models of friction and contact (agreed these, along with deformation, can get fairly complex). Where do the authors see the arbitrary amounts of complexity? Mostly curious here, and I'd like to see a sentence or two describing why the dynamics are complex.
7. Mostly out of idle curiosity, what sparse linear system solver are the authors using to solve for K in eq (6)? At least include this as a footnote or in the Appendix.
8. The authors leverage MuJoCo, which approximates contact dynamics by "softening" them. How "soft" did the authors make the dynamics in their training? How could this impact the ability of the proposed method to move onto real hardware. Do the authors believe their method will be less sensitive to this sim-to-real gap than other learning-based methods? Why?
9. In Section 5.1, the authors state the controller ran at 500 Hz. This is pretty fast. What happens if you slow down the controller? Can you make use of lower-level controllers (like joint-space PID or LQR) to make it easier and run the controller at a lower rate?





**Robotics Focus:**

Highly relevant to robotics but no hardware experiments

**Summary Of Paper:**

The paper presents a method for efficiently learning high degree-of-freedom manipulation tasks by leveraging Koopman operators. Using expert demonstrations, they learn a high-dimensional linear model that can be used to generated reference trajectories that complete the task. This is combined with a learned tracking controller that tracks the generated reference trajectories. The proposed method is evaluated on four complex dexterous manipulation tasks and compared to four baseline reinforcement learning methods. The method performs comparably with state-of-the-art approaches while taking dramatically less time to train.

**Summary Of Recommendation:**

I believe this paper should be accepted to CoRL. The results are compelling, the paper is well-written, and the method offers a different perspective than most of the papers typically seen at this conference. I believe the community will benefit from seeing the results of this paper, which offers a new avenue to explore as we seek to overcome the limitations of traditional neural-network and deep learning methods.

---

### Author Response · Authors · 2023-08-14
**Common response to all reviewers and meta reviewer**

We thank all the reviewers for their thorough and thoughtful review. Their comments have helped us make better arguments and further strengthen our work. We also thank the meta reviewer for finding appropriate reviewers and helping make final decisions.

It appears that all the reviewers generally acknowledge and appreciate the core contributions of our work, recognize our strong empirical results, and perceive the paper as technically clear and well-written. Indeed, our approach stands out within the realm of dexterous manipulation (which is often dominated by deep networks and end-to-end methods) and aims to shed light on alternative approaches that can be equally effective, while significantly improving computationally efficiency and reducing dependence on user expertise.

The reviewers also provided valuable suggestions and raised important questions. We will respond to each reviewer separately to address their individual questions. Below, we summarize changes we had made based on reviewers’ comments. We will attach the revised paper to the official rebuttal to each reviewer.

1. Based on comments from all the reviewers, we have included a new section that explicitly discusses how we use the learned reference dynamics during execution after the learning process. (See Section 4.4 in the revised paper).

2. Following Reviewer k76a’s suggestion, we conducted an additional experiment involving a new BC baseline that learns to map directly from states to actions (as done by some prior works) instead of learning reference dynamics from demonstrations. The results reveal a familiar trend: across all tasks, KODex is drastically more computationally efficient than the new baseline, while performing comparably, if not better, in terms of success rate. (See Appendices M of the revised paper)

3. Following Reviewer RjjK’s suggestions, we examined the inference time to ensure the feasibility of translating KODex to real hardware while meeting necessary control frequency requirements. Our findings indicate that KODex’s inference time is on par with the SOTA baselines and can meet control frequencies as high as 500 Hz. (See Appendices H of the revised paper)

4. In addition to the above, we have made several relatively-minor additions and edits to the write up based on suggestions from the reviewers (indicated by blue text in the revised paper). See also individual responses to reviewers for context and details.

---

### Decision · Program_Chairs · 2023-08-30

**Decision:**

Accept (Oral)

**Comment:**

The paper presents a method for learning manipulation tasks using Koopman operators and leveraging expert demonstrations.

Reviewers appreciated the clarity of presentation, including Koopman operator theory, results showing competitive performance with lower training time,

Reviewers raised concerns about the limitations and future work sections, missing descriptions of the overall pipeline, missing details on learned dynamics, needing more discussion relative to RL/BC/IRL, and lack of real-world demonstration.

The reviewers appreciated the improvements made in the rebuttal but did not raise their assessment further. The paper remains an accept.